# Electron mirror branch: Observational evidence from "historical" AMPTE-IRM and Equator-S measurements

Rudolf A. Treumann[*1] and Wolfgang Baumjohann[2]

[1]International Space Science Institute, Bern, Switzerland
[2]Space Research Institute, Austrian Academy of Sciences, Graz, Austria
[*]On leave from Department of Geophysics and Environmental Sciences, Munich University, Munich, Germany
*Correspondence to*: W. Baumjohann (Wolfgang.Baumjohann@oeaw.ac.at)

**Abstract**. – Based on now "historical" magnetic observations, supported by few available plasma data, and wave spectra from the AMPTE-IRM spacecraft, and on as well "historical" Equator-S high-cadence magnetic field observations of mirror modes in the magnetosheath near the dayside magnetopause, we present observational evidence for a recent theoretical evaluation by Noreen et al. (2017) of the contribution of a *global* (bulk) electron temperature anisotropy to the evolution of mirror modes, giving rise to a separate electron mirror branch. We also refer to related low-frequency lion roars (whistlers) excited by the trapped *resonant* electron component in the high-temperature anisotropic collisionless plasma of the magnetosheath. These old data most probably indicate that signatures of the anisotropic electron effect on mirror modes had indeed been observed already long ago in magnetic and wave data though had not been recognised as such. Unfortunately either poor time resolution or complete lack of plasma data would have inhibited the confirmation of the notoriously required pressure balance in the electron branch for unambiguous confirmation of a separate electron mirror mode. If confirmed by future high-resolution observations (like those provided by the MMS mission), in both cases the large mirror mode amplitudes suggest that mirror modes escape quasilinear saturation, being in a state of weak kinetic plasma turbulence. As a side product, this casts erroneous the frequent claim that the excitation of lion roars (whistlers) would eventually saturate the mirror instability by depleting the bulk temperature anisotropy. Whistlers, excited in mirror modes, just flatten the anisotropy of the small population of resonant electrons responsible for them, without having any effect on the global electron-pressure anisotropy which causes the electron branch and by no means at all on the ion-mirror instability. For the confirmation of both the electron mirror branch and its responsibility for trapping of electrons and resonantly exciting high-frequency whistlers/lion roars, high time- and energy-resolution observations of electrons (as provided for instance by MMS) are required.

*Keywords*: Mirror modes, Electron mirror branch, Magnetosheath turbulence, Lion roars, Weakly kinetic turbulence

## 1 Introduction

Within the past four decades, observations of magnetic mirror modes in the magnetosheath and magnetotail of Earth's magnetosphere, and occasionally also elsewhere, have been ubiquitous (see Tsurutani et al., 2011; Sulem, 2011, for reviews on observation and theory, respectively). They were, however, restricted to the ion mirror mode and the detection of electron-

cyclotron waves (lion roars) which propagate in the whistler band deep inside the magnetic mirror configuration and are caused by trapped resonant anisotropic electrons. (There is a wealth of literature on observations of mirror modes, large-scale electron holes, and lion roars, cf., e.g., Smith and Tsurutani, 1976; Tsurutani et al., 1982; Luehr & Kloecker N, 1987; Treumann et al., 1990; Czaykowska et al., 1998; Zhang et al., 1998; Baumjohann et al., 1999; Maksimovic et al., 2001; Constantinescu et al., 2003; Remya et al., 2014; Breuillard et al., 2018, to cite only the basic original ones, plus a few more recent papers). These observations confirmed their theoretical prediction based on fluid (cf., e.g., Chandrasekhar, 1961; Hasegawa, 1969; Thorne & Tsurutani, 1981; Southwood & Kivelson, 1993; Baumjohann & Treumann, 1996; Treumann & Baumjohann, 1997) and the substantially more elaborated kinetic theory (cf., Pokhotelov et al., 2000, 2002, 2004, and further references in Sulem, 2011), which essentially reproduces the linear fluid results, while including some additional higher order precision terms (like, for instance, finite Larmor radius effects). An attempt of modelling the final state of mirror modes by invoking pressure balance can be found in Constantinescu (2002).

Recently, this theory has been extended to the inclusion of the effect of *anisotropic nonresonant* electrons on the evolution of mirror modes (Noreen et al., 2017, for earlier effects including isotropic thermal electrons, see their reference list) in the linear and quasi-linear regimes. Though in principle rather simple matter, the more interesting finding (when numerically solving the more complicated linear dispersion relation) was that Larmor radius effects are fairly unimportant, while the electrons do indeed substantially contribute to the evolution of mirror modes and in the restriction to quasilinear theory (as the lowest order and therefore believed dominant nonlinear term) also to their quasilinear saturation though, however, in rather different wavenumber and frequency/growth rate regimes.

This finding leads immediately to the question of observation of such effects in the mirror modes in real space, especially to the question *whether signatures of the electron mirror branch had already been present* in any now historical spacecraft observations of mirror modes. Here we suggest that, based on more than three decades old AMPTE-IRM observations in the magnetosheath near the dayside magnetopause and two decades old Eq-S magnetic high resolution observations in the equatorial magnetosheath, both mirror mode branches, the ion as also the electron branch, *most probably* had indeed already been detected in the data though, at that time, the electron branch found by Noreen et al. (2017) only recently in linear theory had remained completely unrecognised. However, the same observations also prove that quasilinear theory as saturation mechanism *does not apply* to real mirror modes, at least not to mirror modes evolving under the conditions of the magnetosheath to large amplitudes where the former measurements had been performed – and probably also not to those observed in the solar wind. All those observations indicate that the mirror mode amplitudes by far exceed those predicted by quasi-linear theory.

## 2 Observations

Figure 1 shows a typical sequence of magnetosheath mirror modes lasting longer than six minutes during an AMPTE-IRM passage on September 21, 1984. The lower panel shows variation of the magnitude of the magnetic field that is caused by the (ion) mirror mode with amplitude $|\delta B| \sim 0.5|B|$. The upper panel is the wave electric power spectrogram. The wavy white line is the electron cyclotron frequency $f_{ce}$ which maps the magnetic field from the lower panel into the frequency

domain. Resonant whistlers (dubbed *lion roars*) emitted in the central mirror mode minima are indicated for two cases. As was shown a decade later (Baumjohann et al., 1999) by visualising their *magnetic wave packet* form from high resolution magnetic field measurements on the Equator-S spacecraft, and thereby directly confirming their electromagnetic nature, do indeed propagate in the whistler band parallel to the magnetic field with central frequency roughly $f_{lr} \sim 0.1 f_{ce}$ of the local central cyclotron frequency. Though barely recognised, these observation were very important for both these reasons. The origin of the other sporadic intense lion roar emissions centred around $f \sim 0.5$-$0.7$ kHz remained unclear. They are not related to the mirror mode minima. They occur at the mirror mode flanks, being of more broadband nature, more temporarily irregular and of higher frequency. For being in the whistler band they require the presence of a trapped resonant anisotropic electron component which is difficult to justify at those locations where they appear. In addition there are irregular high frequency broadband electric signals above $f_{ce}$ reaching up to the local plasma frequency at $f_e \sim 60$-$70$ kHz. Their spiky broadband nature, being independent of the presence of the cyclotron frequency, suggests that they are related to narrow structures or boundaries of which such broadband Fourier spectra are typical (cf., e.g., Dubouloz et al., 1991). The broad unstructured (green) quasi-stationary noise below roughly 2 kHz propagates in the electrostatic ion-acoustic band and is of little interest here as its presence is well-known and is typical for the magnetosheath, being completely independent from the evolution of the mirror mode.

In order to prove that the above sequence of magnetic fluctuations is indeed mirror modes, Figure 2 shows another nearly identical sequence of AMPTE-IRM observations, including plasma data. (Unfortunately, of the former historical sequence no plasma data are available anymore while in the data set used in this figure no wave data have survived.) Maximum time resolution of the magnetic field on AMPTE-IRM was $\sim 30$ ms (32 Hz). We show a 120 s long full resolution excerpt from a long magnetic record. The similarity between the magnetic data in Figures 1 and 2 is striking both in period and amplitude. Four cases are indicated in Figure 2 in order to demonstrate the detectable (at these time resolutions) anti-phase behaviour in the magnetic and plasma data in the magnetic amplitude (or magnetic pressure) panel 1, density in panel 3, and temperature in panel 6. (One may note the logarithmic scale in the temperature.) The anti-correlation is not very well expressed, however, because of the vastly different time resolutions of the magnetic and plasma instruments, stroboscopic and geometrical effects related to the locations and directions of the plasma detector and magnetometer. However the four cases shown give an indication of its presence, which is sufficient for our purposes here. On the other hand it is evident that the available instrumental resolutions inhibited any detection of the notoriously demanded anti-correlations (pressure balance) between fluctuations in the magnetic field pressure and the electron component which are considered the dominant signature of mirror modes[1].

---

[1]In a separate investigation (Treumann & Baumjohann, 2018) applying general thermodynamic arguments we suggested that observed mirror modes obey a substantially more complicated physics than simple pressure balance, linear growth and quasilinear saturation. Being in the final large amplitude thermodynamic equilibrium state, the mirror mode, even though identified by approximate pressure balance as a good indicator of mirror actions, requires the contribution of diamagnetic currents flowing on the boundary surfaces, i.e. the magnetic stresses must be included. In the evolution of mirror modes to large amplitudes this leads to correlations between the mirror trapped particles which is taken care of in a correlation length and sets an important condition for the decay of the mirror mode into a chain of separate large-amplitude bubbles.

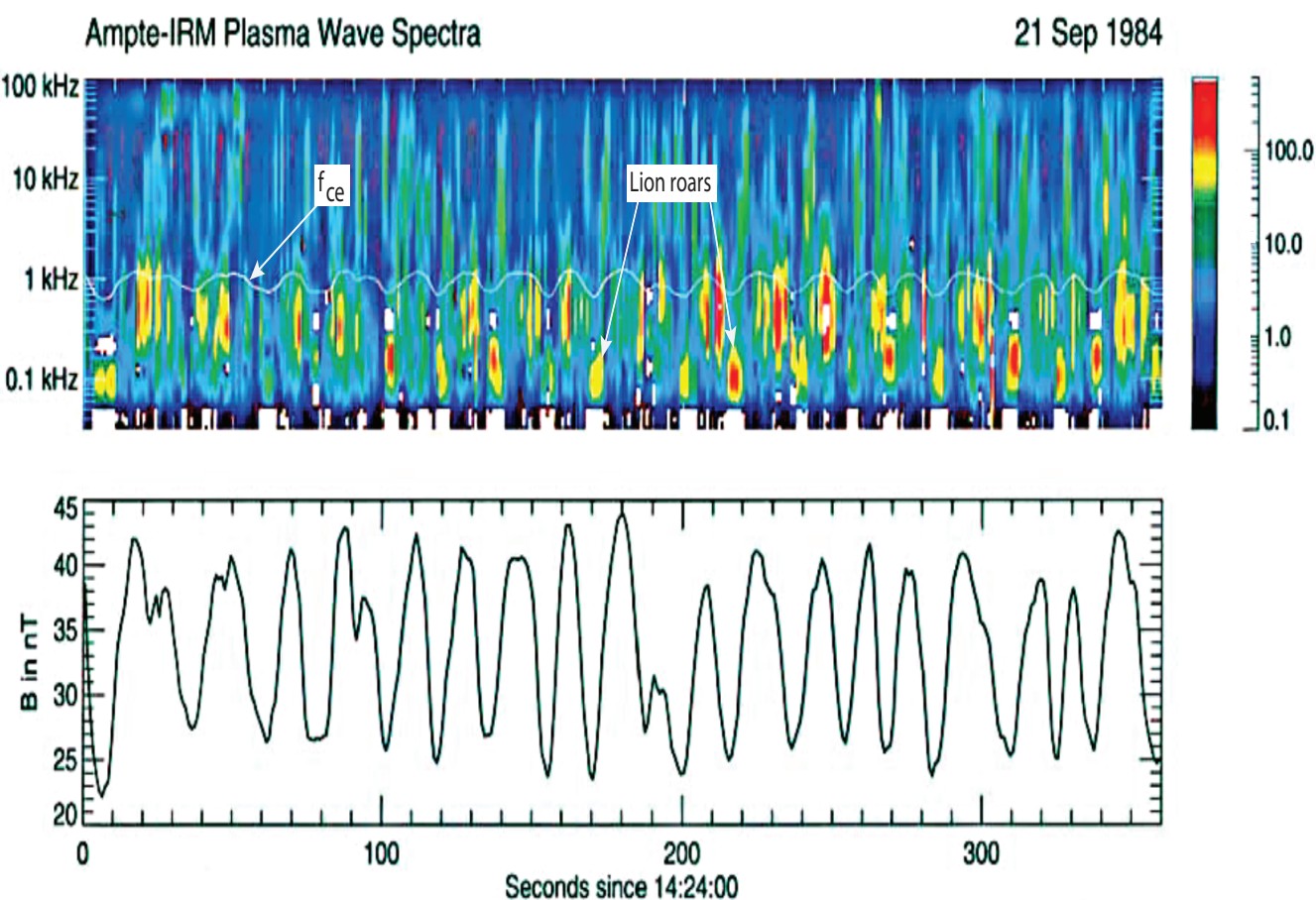

**Figure 1.** AMPTE-IRM observations of mirror modes in the magnetosheath and related plasma wave power spectra (see the colour bar on the right for relative log-scale intensities). Indicated are the electron cyclotron frequency $f_{ce}$ (white trace mimicking the mirror mode magnetic field in the lower panel), and lion roar emissions in the mirror mode minima (cf., e.g., Baumjohann et al., 1999) at frequency $f_{lr} \sim 0.1 f_{ce}$ (after Treumann et al., 2004a). The higher frequency sporadic emissions below the electron cyclotron frequency are related to the flanks of the ion mirror mode and are interpreted as high frequency lion roars (high frequency whistlers) caused by resonant electrons trapped in the electron mirror branch oscillations which develop in the ion mirror mode and are caused by the bulk electron temperature anisotropy of the magnetosheath plasma. The weak broadband signals extending in frequency above and beyond $f_{ce}$ are broadband electrostatic noise (most clearly seen, for instance, around 150 s). They are most probably related to the steep trapped electron-plasma boundaries which locally form when the electron mirror branch evolves and thus also correlate with high frequency lion roars. In any case, extension of lion roars beyond $f_{ce}$ into the purely electrostatic frequency range is clear indication of the presence of steep electron plasma gradients (cf., e.g., Dubouloz et al., 1991, for the general arguments). High frequency electric emissions centred around $> 3$ kHz and above are of different nature. In this range above plasma and upper hybrid frequencies they are mostly sporadic electromagnetic noise in the magnetosheath plasma turbulence.

We will argue that the broadband sporadic nature of the unidentified emissions, their relation to the flanks of the ion mirror mode, and intensification below the local cyclotron frequency suggests that they are the signatures of electron-mirror branch structures which are superimposed on the ion-mirror branch which dominates the gross behaviour of the magnetic field.

For this purpose we refer to a rare observation by the Eq-S spacecraft at the high magnetic sampling rate of 128 Hz which is reproduced in Figure 3. Unfortunately, as had already been noted (Baumjohann et al., 1999), no plasma measurements were available due to failure of the plasma instrument. However, even if the plasma instrument would have worked properly, the resolution of optimum only $\sim 3$ s spacecraft spin and thus comparable to the plasma resolution of AMPTE-IRM ($\sim 4$ s) would anyway not have been sufficient for resolving the electron structures in the plasma data and establishing/confirming any pressure balance between the electron fluid (not the resonant electrons responsible for the lion roars) and magnetic field, as was done with the AMPTE observations for the ion-mirror modes. What concerns the identification of the large amplitude magnetic oscillations as genuine ion-mirror modes, even though no plasma measurements were available on Eq-S, the reader is referred to Lucek et al. (1999a, b) who analysed the whole sequence of magnetic oscillations measured by Eq-S of which our short high-resolution example is just a sample selection. Figure 3 shows this high-resolution record (used by Baumjohann et al., 1999, in the investigation of lion roars) of the magnetic field magnitude from this data pool of Eq-S. Just two ion-mirror oscillations of the cycle analysed in Lucek et al. (1999a, b) and used in that paper are shown here. One may note that their period and amplitude is of the same order as in the case of the AMPTE observations which were approximately at a similar location in the magnetosheath, thereby providing additional independent confidence for them as being ion mirror modes. One observes the general evolution of the magnetic field which is reflected in the slight asymmetries of the structures which pass over the spacecraft. These might be caused by temporal evolution of the mode or also by crossing the spatially densely packed mirror mode oscillations (which form kind of a "magnetic crystal texture" of magnetic bottles on the plasma background in the magnetosheath, as sketched in Treumann & Baumjohann, 1997, pages 57-58) by the spacecraft under an angle. The maximum of the magnetic field in this case is $\sim 30$ nT with a $|\delta B|/B \sim 50\%$ amplitude oscillation, almost identical to what AMPTE-IRM had observed. The very small-amplitude high-frequency fluctuations of the field in the field minima belong to the lion roars mentioned above and have been investigated in detail Baumjohann et al. (1999). In the left hand field minimum at the bottom of the ion-mirror oscillation the small local maximum of the magnetic field (like on the bottom of a wine bottle) can be recognised to which Baumjohann et al. (1999) refer as an "unexplained" structure which, as shown here, turns out to be the signature of the electron branch caused by the bulk temperature/pressure anisotropy of the ion-mirror trapped electron component.

The importance of the investigations by Baumjohann et al. (1999) for the physics of mirror modes and lion roars lies in the fact that, by measuring the *finite magnetic amplitude waveform and nonlinear wave-packet form* of the lion roar whistler fluctuations $\delta \boldsymbol{B}$, frequency, and polarisation, they demonstrated unambiguously that those low frequency lion roars were indeed electromagnetic waves propagating on the whistler mode. Their spectral analysis proved, in addition, that these waves propagated in large-amplitude spatially confined packets of nonlinear whistler waves, a most important achievement as it demonstrated their nonlinear evolution. Until then all conclusions concerning lion roar whistlers were based solely on spectral wave electric field measurements $\delta \boldsymbol{E}$ like those in the AMPTE-IRM data of Figure 1 combined with secondary arguments.

Earlier magnetic wave instruments than that on Eq-S did not provide any magnetic wave measurements in this low frequency range. It was, moreover, shown in that work that the occurrence of lion roars was related to the presence of a residual *weak resonant anisotropy* in the electrons left over after quasilinear saturation of the lion roars. This resonant particle anisotropy was independent on the global pressure anisotropy. The latter remains unaffected by the excitation of lion roars or whistlers.[2]

Three kinds of magnetic variations are visible in this figure. Firstly, we have the large amplitude ion-mirror mode oscillations of which only two periods are shown. Secondly, superimposed on these are the spiky small amplitude excursions from the ion mirror shape which form small peaks and valleys everywhere on the flanks, maxima and minima in rather irregular or at the best quasi-regular sequence. Thirdly and finally, there are very small amplitude oscillations which, as far as the instrument can resolve them, accumulate mainly in relation to the former medium frequency and medium amplitude magnetic modulations. The interesting feature in this high-resolution recording of the magnetic field are these medium frequency medium amplitude tooth-like oscillations of the magnetic field in the flanks, in the maxima and also in the minima of the ion-mirror mode. In this respect the high resolution magnetic field in this figure is quite different from the apparently smooth four-times lower resolution and less sensitive course of the mirror field in the AMPTE-IRM magnetosheath observations of Figs. 1 and 2. We repeat that these Eq-S chains of magnetic modulations had indeed been identified as mirror modes by Lucek et al. (1999a), even though no plasma data were available. We also repeat that, even if the plasma instrument on Eq-S would have worked properly, its time resolution of $\sim 3$ s would not have been sufficient to resolve any anti-correlation between the magnetic and plasma pressures in the oscillations which we here and below identify as electron mirror modes. Measuring this anti-correlation would have required a substantially or even much higher time resolution than the spin resolution, which was only available at the times of those spacecraft.

In order to infer about the nature of these oscillations we refer to the period of the (ion) mirror mode. This can be read from the figure to be roughly $\tau_{im} \sim 30$ s, corresponding to a frequency of $f_{im} \sim 0.03$ Hz. The tooth-like oscillations, for instance at the first steep increase of the magnetic field, have a time period of $\tau_{em} \sim 2 - 4$ s (or frequency $f_{em} \sim 0.3$ Hz), roughly a factor of ten shorter than the ion-mirror mode. In addition to their steep magnetic boundaries, these structures also exhibit superimposed small frequency oscillations (appearing as broadenings of the magnetic trace as shown in Figure 4), which are also present in the modulated maxima of the mirror mode. Since the latter belong to magnetic fluctuations, it is reasonable to assume that they are simply a different kind of lion roars caused by electrons trapped in the local minima of the higher frequency-shorter wavelength modulations. Thus their centre frequency should be higher than the lion roar frequency in the ion-mode minima. This suggests identification with the higher frequency spectral features observed by AMPTE-IRM, while the weak broadband features in the wave spectra may be related to the steep magnetic and plasma (pressure) boundaries of the modulations.

---

[2]It is worth noting that the (almost completely ignored) Baumjohann et al. (1999) paper should have deserved recognition for its important findings: Measurement of waveform of lion roars at bottom of ion mirror troughs, proof of their large amplitudes and nonlinear wave-packet structure, identification of bulk electron temperature anisotropy, and range of resonant electron energy responsible for the excitation of the observed lion roars.

## 3 Electron mirror branch

If this is the case then it is suggestive to identify the small amplitude modulations in the magnetic field seen by Eq-S and in the wave spectra by AMPTE-IRM with the electron mirror mode which was theoretically predicted (Noreen et al., 2017). These authors put emphasis on the quasilinear evolution of the pure electron (ion) and mixed (electron-ion) mirror modes to numerically show for a number of cases how the normalised magnetic and plasma energy densities evolve and saturate. For our purposes it suffices to consider the mixed linear state, because it is clear from the data in Fig. 3 and as a consequence also in the spectrum of Fig. 1 that the dominant magnetic (and also plasma) structure is the ion mirror mode while the electron mirror mode just produces some modification. Clearly the ion mirror mode is a large-scale perturbation on which the quasilinear contribution of the electron mirror mode does not change very much.

For our purposes we need only the simplified purely growing linear growth rate (normalised to the ion-cyclotron frequency $\omega_{ci} = eB/m_i$) for small ion and electron arguments

$$\frac{\gamma(\boldsymbol{k})}{\omega_{ci}} \approx \frac{k_\parallel \lambda_i}{A_i + 1} \sqrt{\frac{\beta_{\parallel i}}{\pi}} \left[ A_i + \sqrt{\frac{T_{e\perp}}{T_{i\perp}}} A_e - \frac{k^2}{k_\perp^2 \beta_{i\perp}} \right] \tag{1}$$

with $\lambda_i = c/\omega_i$ the ion inertial length, $\omega_i^2 = e^2 N/\epsilon_0 m_i$ square of ion plasma frequency, $A_j = (T_\perp/T_\parallel)_j - 1 > 0$ the temperature anisotropy of species $j = i, e$, and $\beta_{\parallel i} = 2\mu_0 N T_{i\parallel}/B^2$ the ratio of parallel ion thermal and magnetic energy densities (cf., Noreen et al., 2017). Wave growth occurs for positive bracket which provides angular dependent thresholds. The threshold condition for instability can be written in terms of the magnetic field as

$$B < B_{crit} \approx \sqrt{2\mu_0 N T_{i\perp}} \left( A_i + \sqrt{\frac{T_{e\perp}}{T_{i\perp}}} A_e \right)^{\frac{1}{2}} |\sin\theta| \tag{2}$$

where $\theta = \sin^{-1}(k_\perp/k)$, and the approximate sign refers to the simplifications made in writing the simplified dispersion relation. Once the local magnetic field drops *below* this threshold value, instability will necessarily set on. Such a critical value $B_{crit}$ exists for all combinations of anisotropies which leave the sum under the root positive. It has a deeper physical meaning (Treumann et al., 2004a) corresponding to a classical Meissner effect in superconductivity. This threshold relates to the critical mirror mode angle of Kivelson & Southwood (1996). It sets an angular dependent upper limit on the critical magnetic field which vanishes for parallel and maximises for perpendicular propagation. In both these cases, however, no instability can arise, as follows from the growth rate, and the instability, as is well known, is oblique. The growth rate obtained from the simplified dispersion relation maximises formally at a maximum angle $\theta_{max}$ with

$$\sin^2\theta_{max} \approx \frac{1}{2a}\left\{ \sqrt{1 + 8a} - 1 \right\}$$

$$1 < a \equiv \beta_{i\perp} \left[ A_i + \sqrt{\frac{T_{e\perp}}{T_{i\perp}}} A_e \right] \tag{3}$$

For $a = 2$, for instance, this yields $\theta_{max} \approx 45°$ in accord with the exact numerical calculation for the ion mode based on the full dispersion relation (Noreen et al., 2017). No maximum exists for $a \leq 1$. Folding the growth rate with the threshold condition

and maximising yields the optimum unstable range. This gives a third order equation for $x = \cos 2\theta_{opt}$:

$$x^3 - \left(1 - \frac{2}{a}\right)x \pm \frac{1}{a} = 0 \quad \text{for} \quad \begin{cases} \theta_{opt}^{(1)} & < \quad \pi/4 \\ \theta_{opt}^{(2)} & > \quad \pi/4 \end{cases}$$

depending on the sign of the last term. Expanding near parallel propagation $x - \xi \approx 1$ for $a \gtrsim 2$ and very roughly keeping only the linear term in $\xi$, gives two solutions which approximately fix the range $\theta_{opt}^{(1)} < \theta < \theta_{opt}^{(2)}$ of maximum growth for the ion mode, corresponding to angles $\theta_{opt}^{(1)} \approx \frac{1}{2}\cos^{-1}\left(\frac{1}{2}\right) \approx 30°$ and $\theta_{opt}^{(2)} \approx \frac{1}{2}\cos^{-1}\left(\frac{1}{6}\right) \approx 50°$. These considerations apply to the simplified dispersion relation used in this paper. They don't discriminate between the roles of ion and electron gyro radii. This distinction is contained in the full dispersion relation on which the numerical solution (Noreen et al., 2017) is based, thereby leading to the numerically obtained precise angular ranges for the two branches of the mirror instability to which we refer below.

The pure electron effect which applies to the electron branch is obtained for isotropic ions $A_i = 0$ and $T_{i\parallel} = T_{i\perp} \equiv T_i$. On the electron gyroradius scale the ions are unmagnetised, yielding

$$\frac{\gamma_e(\boldsymbol{k})}{\omega_{ci}} \approx k_\parallel \lambda_i \sqrt{\frac{\beta_{e\parallel}}{\pi}} \left[A_e - \frac{k^2}{k_\perp^2 \beta_{e\perp}}\right]\sqrt{A_e + 1}$$
$$B_{crit,e} \approx \sqrt{2\mu_0 T_{e\perp} A_e}\left|\sin\theta\right|$$

The perpendicular mirror scale and critical threshold magnetic field are determined by the electron anisotropy $A_e$ and perpendicular electron thermal energy ratio $\beta_{e\perp}$.

Effectively, the electron-mirror branch remains to be a separate branch on the ion-mirror instability with parallel scale determined by the ion inertia, while its perpendicular scale and critical excitation threshold are prescribed by the electron dynamics. The perpendicular scale of the electron branch is much shorter than the ion scale, while the threshold depends only on the electron temperature and anisotropy. Writing the growth rate in pure electron quantities, one has for the isotropic-ion electron branch

$$\begin{aligned} \frac{\gamma_e(\boldsymbol{k})}{\omega_{ce}} &\approx \sqrt{\frac{\beta_{e\parallel}}{\pi}}\left[A_e - \frac{k^2 e^{\zeta_e}}{k_\perp^2 \beta_{e\perp}[1 - \zeta_e/2]}\right]\frac{k_\parallel \lambda_e}{D} \\ D &\equiv 1 + \sqrt{\frac{m_i}{m_e}\frac{T_{e\perp}}{T_i}}\frac{\exp[-(\zeta_i - \zeta_e)]}{(A_e + 1)^2} \\ \zeta_e &= k_\perp^2 \lambda_e^2 \beta_{e\perp} \ll 1 \end{aligned} \qquad (4)$$

The ion term $D$ in the nominator acts stabilising on the electron branch though, because of the large square of the ion gyroradius $\zeta_i/\zeta_e = (m_i/m_e)T_{e\perp}/T_i \gg 1$, the additional term in $D$ is exponentially reduced. The electron inertial scale $\lambda_e$ enters to replace $\lambda_i$. This expression shows the similarity between the ion and electron branches, however with different scales and an increased threshold for the electron branch. The electron-branch growth rate depends on the ion temperature. When $T_i$ becomes large, the growth of the electron contribution will be suppressed. In contrast to the magnetosheath this should be the case, for instance, in Earth's magnetotail plasma sheet where one has $T_i \sim 10 T_e$. Presumably any mirror modes which evolve there will be void of an electron branch.

In the last expressions the direction of anisotropy is with respect to the local magnetic field as the electrons experience it. It can be rather different from that of the main branch of the ion mirror mode. These effects are still of first order, being independent of any finite Larmor-radius contributions (Pokhotelov et al., 2004) which occur in higher order approximation, having been shown (Noreen et al., 2017) to be of minor importance. This might not be the last word, because these authors investigate just the linear and quasilinear evolution of mirror modes. Below we comment on this important point.

It is clear from here that, based on our arguments and the numerical calculations (Noreen et al., 2017, see their Fig. 1), the two branches of the mirror mode grow in separate regions of wavenumber space $k_\parallel, k_\perp$, where the indices refer to the directions parallel and perpendicular to the local magnetic field, i.e. in the ion mirror mode to the average ambient magnetic field which is modulated by the mirror mode, on the electron branch the local magnetic field at the location where the electron mirror bubble evolves. This main and well expected effect in the combined electron-ion growth rate found by numerically solving the complete non-simplified growth rate (Noreen et al., 2017, their Eq. 4) is that, because of the different gyration scales of electrons and protons, the growth rate exhibits its two separate branches and maximises at different angles for the two branches. Since in the linear state the different modes extract their energy from the component of particles to which they belong, the two modes grow separately but not independently because of their inertial coupling and the modification of the local magnetic field by the ion branch. It is this local field and thus mostly reduced field which the electrons feel. Only in the quasilinear state an exchange in energy takes place (as shown by Noreen et al., 2017).

Their linear numerical calculations demonstrate clearly that in absolute numbers the electron branch growth rate (measured in ion cyclotron frequencies) is about an order of magnitude larger than that of the ion mirror mode. It grows faster and, as a consequence, saturates readily, such that one expects it to be of comparably small final amplitude. The ion mirror mode growth rate maximises at $k_{\parallel i} \approx k_{\perp i}$, which corresponds to an angle of $\sim 45°$ with respect to the ambient magnetic field direction, while the electron branch mode is nearly perpendicular $k_{\parallel e} \approx 0.1 k_{\perp e}$, i.e. it is of much shorter perpendicular than parallel wavelength. On the other hand, the maximum unstable parallel wavelengths are comparable, $k_{\parallel e} \approx 3 k_{\parallel i}$, while the maximum unstable perpendicular wavelengths are different: $k_{\perp e}/k_{\perp i} \approx 20 - 30$ for the parameters investigated (Noreen et al., 2017). The electron mirror branch structure is elongated essentially parallel to the local field, while the ion mirror branch is oblique to the ambient field. Whereas the ion mirror mode tends to form large magnetic bubbles, the electron mirror mode forms narrow long bottles on the structure given by the ion mirror mode. We may assume that this behaviour will not be very different for other parameter choices than those used in the numerical calculation, as it is just what one would expect: the electron mirror mode being somewhat longer in parallel wavelengths and substantially shorter in perpendicular wavelengths than the ion mirror mode, an effect of the vastly different gyroradii.

## 4 Discussion

With this information at hand we consult the high resolution Eq-S observations in Fig. 3. This figure suggests that Eq-S was crossed by the chain of mirror structures *almost* in the perpendicular direction. Comparing the times of crossing the large-scale ion mirror mode and the well expressed small-scale structures on the flank of the first rising boundary, we infer that the ratio

of wavelengths between the long and short structures is of the order of roughly a factor of $\sim 10$. Though this is not exactly the above value for this ratio in the perpendicular direction, it is pretty close to the expectation that the small scale structure is caused by the electron component thus representing the electron mirror mode (indicated already in Sect. 2 by the subscript *em*).

*Reference to* Noreen et al.'s (2017) *linear theory*: There are a number of shorter structures of smaller amplitudes visible the use of which would come closer to the canonical scales obtained from the numerical calculation of the maximum growth rates (Noreen et al., 2017). However there are many reasons for staying with this result. The first would be the choice of the parameters chosen by Noreen et al. (2017) for their linear and quasilinear calculation. Another and more important one is that even for those parameters the spread of the domain of maximum growth of both the ion and electron mirror modes (cf. Fig.1 in

Noreen et al., 2017) is sufficiently large for fitting the spread in the measurements. We may apply the half-maximum condition to the numerical calculation for exponential growth. Inspecting the growth rate plot, the wave power is one fourth of its value at maximum growth for wave numbers $k_\perp$ with growth rate $\gamma(k_\perp) \sim 0.3\gamma_{max}$. With this value the figure implies a spread in $k_\perp$ for the electron mode of $\Delta k_\perp \lambda_i \sim 5$, large enough for covering a sufficiently broad interval of electron mirror wavelengths to account for the time or wavelength spread in our Figure 3. Finally, just to mention it, it is not known from the observations,

in which direction precisely the electron mirror mode would propagate relative to the ion mirror mode. Theory suggests propagation nearly parallel to the *local mean* magnetic field. This might become fixed with MMS observations. Hence, the above conclusion, though still imprecise, should suffice as evidence for the observation of both electron and ion mirror modes in the magnetosheath by Eq-S with both modes acting simultaneously in tandem. Unfortunately, as noted above, no plasma observations were available such that we are not in the position to provide a more sophisticated investigation, in particular the

wanted pressure balance between the electron mirror field and electron pressure implicitly contained in the theory. But even if the Eq-S plasma instrument would have worked properly, its poor time resolution of $\sim 3\,\mathrm{s}$ would not have been sufficient for demonstration of any pressure balance.

    Since electron mirror branches have so far not been reported, there might be substantial reservation accepting that they are indeed present. This makes it necessary to provide additional arguments both theoretical and experimental.

*Theoretical arguments for an electron mirror branch*: The short-scale modulations of the magnetic field modulus seen in both the AMPTE-IRM and Eq-S recordings are well below the ion-gyroradius scale. Hence on these scales ions are nonmagnetic and thus do not contribute to magnetic oscillations while electrostatic waves do anyway not contribute. From the ion point of view the only wave which could be made responsible is electromagnetic ion-cyclotron waves (electromagnetic ion-Bernstein modes) which at frequencies above the ion cyclotron frequency have rather weak amplitudes. Moreover, if present, they should

be seen in the spectra as chains of harmonics. This is not the case. The other possibility would be Weibel modes which have wavelengths the order of the ion-inertial scale but do barely grow in non-zero magnetic fields. When growing they have finite frequency near the ion cyclotron frequency and very weak amplitudes. Moreover, they require the presence of narrow antiparallel ion beams whose origin would not be known.

    Since ion modes are probably out, we are left with electromagnetic electron modes, viz. whistlers/electron-Alfvén waves.

The electron-mirror branch propagates on this mode as a long-wavelength oblique whistler with $k_\perp > k_\parallel$. (For the angular

range see our above discussion.) The other possibility is electromagnetic drift waves propagating perpendicular to the magnetic field and the gradients of density and field. Their magnetic component is caused by the diamagnetic currents flowing in these waves and is therefore parallel to the ambient field. Such waves cannot be excluded, which is in contrast to the above mentioned modes. However, these modes are secondarily excited while the electron branch, as shown by Noreen et al. (2017) is

a linear first order mode and should therefore grow faster and stronger. Nevertheless, the possibility remains that electron-drift waves are observed. The only argument against them is that in several cases they appear in the magnetic minima and maxima where the magnetic and density gradients vanish and they could only be present when propagating into those regions. Still in all those cases the unanswered question remains why those waves have comparably large amplitudes and do not form sinusoidal wave chains. Thus denying the existence of the electron mirror branch as theoretically inferred in Noreen et al. (2017) simply

shifts the explanation of the observed magnetic structures into the direction of some other unexplained effect. This is unsatisfactory also from the point of view that the simple argument these structures would be incidentally caused being "nothing but" fluctuations does not work. For such stochastic fluctuations their amplitudes are by orders of magnitude too large. Thermal fluctuations (Yoon & López, 2017) are invisible on the magnetic traces both in AMPTE-IRM and Eq-S.

*Experimental arguments*: It also makes sense to provide some experimental arguments concerning the observation of what

we call high-frequency lion roars (whistlers) in the flanks and on top of the mirror modes, even though by now, after the recent publication of a detailed in-depth investigation (Ahmadi et al., 2018, see the *Note added in final revision*) of those wave modes based on MMS data, this is not anymore required.

Both AMPTE-IRM and Eq-S had their spin axes perpendicular to the ecliptic. On AMPTE-IRM the electric wave antenna was perpendicular to the spin axis, the magnetometer was perpendicular to it though oblique to the field. The average field (Fig.

2, panel 2) was about perpendicular to the Sun-Earth line, parallel to the shock and magnetopause. Hence the antenna measured the transverse electric field of a parallel propagating electromagnetic mode modulated at twice the spin frequency, too slow for resolving the modulation during one short passage across any of the whistler sources. This cannot be directly resolved in Fig. 1 though modulations of the spectral intensity can be seen but are obscured by the stroboscopic effect of the rotating antenna and the occurrence of the electron mirror structures. In the particular range of frequencies below the electron cyclotron

frequency in Fig. 1 one should, however not expect any other waves except the weak about stationary electrostatic ion-acoustic noise (Rodriguez & Gurnett, 1975) mentioned earlier. This noise propagates parallel and oblique but drops out perpendicular to the magnetic field. The occurrence of the high-frequency intense signals coinciding with some of the dropouts of ion-sound indicate that then the electric field measured was more or less strictly perpendicular to the magnetic field, thus being in the electromagnetic parallel propagating whistler mode. This should be sufficient argument here for parallel propagating whistlers

wherever high-frequency lion roars were observed.

Eq-S did not carry any wave instrumentation. Hence the only signatures of lion roars at higher frequencies are the broadenings of the magnetic traces. The wave forms and spectra of Baumjohann et al. (1999) proved that those waves occurred in sharply confined large amplitude nonlinear wave packets. Such spatial packets cause two kinds of signals, the noted broadband high-frequency electric signals (Dubouloz et al., 1991) clearly seen in Fig. 1 exceeding the electron cyclotron frequency and,

when time-averaging their magnetic components, broadenings of the magnetic trace. Not being simple sinusoidal oscillations,

they contribute an average non-vanishing rms amplitude to the ambient field at the spatial location of the nonlinear wave packets which causes the broadening of the magnetic traces. This is both seen in Fig. 3 and in Fig. 4 respectively.[3]

## 5 Conclusions

Accepting that Eq-S indeed observed both branches in the magnetosheath, reference to Fig. 3 further suggests that, as suspected, the intense higher-frequency unidentified emissions beneath the electron cyclotron frequency in Fig. 1 represent the equivalent to lion roars in the ion mirror mode though now on the electron mirror branch. Recently Breuillard et al. (2018, see their Fig. 1) in analysing MMS electron and wave data observed high frequency whistler waves at the edges of mirror mode packets in relation to a perpendicular anisotropy in the electron temperature (see Fig. 5 in Breuillard et al., 2018) which, in the light of our claim, provides another indication for the presence of an electron mirror branch. In fact, it would be most interesting to check whether the high sampling rate of 33.3 Hz on MMS suffices for detecting a pressure balance between electron mirror-scale magnetic oscillations and electron pressure. This should decide whether the electron branch is an own bulk-electron-anisotropy driven branch of the mirror mode or just a magnetic oscillation which is not in pressure balance.

The perpendicular extension of an electron mirror branch, though being shorter than the ion inertial length $\lambda_i = c/\omega_i$ is, for a magnetic field of the order of $B \approx 25$ nT as in the measurements of Eq-S, still substantially larger than the electron gyroradius, which is a general condition for the electron mirror branch to exist and grow on magnetised globally anisotropic electrons. Electrons trapped inside those structures on the electron mirror branch will, by the same reasoning (cf., e.g., Thorne & Tsurutani, 1981; Tsurutani et al., 1982, 2011; Baumjohann et al., 1999, and others), be capable of exciting the whistler instability and thus produce high frequency lion roars, still below the *local* electron cyclotron frequency, which in this case would be around $f \sim 0.5 - 0.7$ kHz, in reasonable agreement with the majority of high intensity emissions below the local electron cyclotron frequency in Fig. 1. (It also corresponds to the MMS observations reported by Breuillard et al., 2018). These are found to coincide with the walls and maxima of the main ion mirror structures, and in some cases evolve even on top of the maxima (see the cases indicated in Fig. 3) in the magnetic field strength. Quasilinear quenching of the anisotropy to a low level is no argument against the presence of lion roars. A low level of bulk temperature/pressure anisotropy will always be retained even quasilinearly (cf., e.g., Treumann & Baumjohann, 1997). Excitation of whistler waves will occur if an anisotropic *resonant* electron component is present.

Hence the higher-frequency waves related to the mirror structures are presumably caused by those anisotropic resonant electrons which may become trapped in the secondary electron mirror branch structures (Breuillard et al., 2018, in their Fig.5, report a perpendicular electron anisotropy in relation to the observation of high frequency waves) which grow on the background magnetic field and plasma structure of the ion mirror mode. Considering that mirror modes trap electrons and that there is plenty of reason for the trapped electrons to evolve temperature anisotropies as well as a higher energy resonant electron component, this is quite a natural conclusion. On the other hand, the weak *broadband electric* emissions exceeding the ambient cyclotron frequency and being irregularly related to the ion mirror structures then presumably result from steep plasma

---

[3]For a recent proof of their presence the reader is directed to Ahmadi et al. (2018).

boundaries on the shorter scale electron mirror branch structures, i.e. from their trapped electron component which should be responsible for the maintenance of the (currently otherwise undetectable) local total pressure balance in them, when traversing the spacecraft at the high flow speeds in the magnetosheath. Generation of such broadband signals in steep electron gradients are well known from theory and observation of ion and electron holes.

5    At this occasion a remark on the *saturation* of the mirror mode is in place. It is sometimes claimed that quasilinear saturation, because of the exponential self-quenching of the growth rate, readily limits the achievable amplitude of a linear instability to rather low values of at most few percent or less. This behaviour is clearly seen in the numerical calculations of the quasilinear mirror saturation level (cf., Noreen et al., 2017, e.g. their Fig. 2, where the magnetic amplitude settles at $\langle \delta B \rangle / B \lesssim 0.2\%$ of the main field). For the mirror mode it results in quasi-linear depletion of the global temperature anisotropy (cf., e.g., Treumann, 10    1997; Noreen et al., 2017). The same argument applies to the electron whistler instability (leading to lion roars) which (since Kennel & Petschek, 1966; Vedenov et al., 1961) is known to quench the responsible *resonant* electron temperature anisotropy until reaching a balance between a rudimentary level of anisotropy/resonant particle flux and moderate wave intensity, an argument that also applies to the generation of lion roars.

However, there is an important and striking difference between the two saturation processes in mirror modes, whether on the 15    ion or electron mirror branches, and quasilinear saturation of whistlers. This difference frequently leads to misconceptions. Mirror modes result from macroscopic (fluid) instability the source of which is the *global pressure / temperature* anisotropy of the bulk of the particle distribution. This anisotropy provides the free energy for the mirror instability. Whistlers and ion-cyclotron waves, on the other hand, take their energy from the presence of a population of anisotropic *resonant* particles. Depletion of the latter by quasilinear saturation of the whistler instability has little effect on the *global* temperature anisotropy which 20    drives mirror modes unstable; it partially quenches the emission of whistlers (or in a similar way also ion-cyclotron waves). Since, however, quasilinear saturation never completely depletes the initial resonant anisotropy, some amount of anisotropy will even in this case remain (the reader may consult the related simulations of Sydora et al., 2007, which apply to the whistler case). Such a rudimentary anisotropy which is located in velocity space around the resonant population may, if large enough, nonetheless contribute to a global anisotropy which then affects the evolution of the mirror instability.

25    Though violent quasilinear suppression of the mirror instability and saturation at a low quasilinear level seem reasonable, they contradict the observation of the $|\delta B|/B \sim 50\%$ amplitudes reported here and elsewhere. Restriction to quasilinear saturation ignores higher-order nonlinear interactions. It is well known that in many cases these additional weakly-turbulent effects undermine quasilinear saturation as, for instance, occurs in one of the most simple and fundamental instabilities, the gentle-beam-plasma interaction (the reader may consult Yoon, 2018, for a most recent and exhaustive review of this basic plasma 30    instability which serves as a paradigm for all instabilities in a hot collisionless plasma). In fully developed weak plasma turbulence (cf., e.g., Sagdeev & Galeev, 1969; Davidson, 1972; Tsytovich, 1977, for the basic theory), various mode couplings and higher-order wave-particle interactions *erase* the process of flat straightforward quasilinear stabilisation. Under weak turbulence, the instability evolves through various stages of growth until finally reaching a quasi-stationary turbulent equilibrium very different from being quasilinear. In the particle picture, this equilibrium state is described by a generalised Lorentzian 35    thermodynamics (Treumann, 1999a, b; Treumann et al., 2004b; Treumann & Jaroschek, 2008; Treumann & Baumjohann,

2014a) resulting in the generation of power-law tails on the distribution function which have been observed since decades in all space plasmas.

The weakly-turbulent generation of the quasi-stationary electron distribution, the so-called $\kappa$-distribution (which is related to Tsallis' $q$-statistics, cf., Tsallis, 1988; Gell-Mann & Tsallis, 2004, see also the review by Livadiotis 2018, and the list of references to $q$ statistics therein; the relation between $\kappa$ and $q$ was given first in Treumann 1997a), was anticipated in an electron-photon-bath calculation by Hasegawa et al. (1985), but the rigorous weak-turbulence theory, in this case providing an analytical expression for the power $\kappa$ as function of the turbulent wave power, was given first by P.H.Yoon for a thermal electron plasma with stationary ions under weakly turbulent electrostatic interactions, including spontaneous emission of Langmuir waves, induced emission and absorption (Landau damping) (see Yoon, 2014, and the reference therein).

Similar electromagnetic interactions take place in weak magnetic turbulence (cf. Yoon, 2007; Yoon & Fang, 2007, for an attempt of formulating a weakly-turbulent theory of low-frequency magnetic turbulence). At high particle energies this power law becomes exponentially truncated when other effects like spontaneous reconnection (Treumann & Baumjohann, 2015) in magnetically turbulent plasmas or particle-particle collisions ultimately come into play when the life time becomes comparable to the collision time (Yoon, 2014; Treumann & Baumjohann, 2014b). Until this final quasi-stationary state is reached, the instability grows steadily in different steps, thereby assuming substantially larger amplitudes than predicted by quasilinear theory.

For magnetic mirror modes no weakly turbulent theory has so far been developed yet. The observed $|\delta B|/B \sim 50\,\%$ amplitudes of the ion-mirror modes in Figures 1 and 2, and the comparably large amplitudes of the inferred electron-mirror branch oscillations recognised in Figure 3 are much larger than quasilinearly expected (Noreen et al., 2017). The electron-mirror branch amplitude inferred from Figure 3 amounts to $\langle \delta B \rangle/B \sim 5\,\%$, one order of magnitude less than for the ion mode though as well still much larger than quasilinearly predicted. Such large amplitudes suggest that both branches, the ion-mirror as well as the electron-mirror branch, do in fact *not saturate quasilinearly*. Rather they are in their weakly turbulent quasi-stationary state. Presumably they have not yet reached their final state of dissipation and at least not that of their ultimate dissipative or even collisional destruction. Both being anyway irrelevant at the plasma flow times in the magnetosheath.

The question for a weak turbulence theory of mirror modes has so far not been brought up, at least not to our knowledge. It should be developed along the lines which have been formally prescribed by Yoon (2007) for low frequency isotropic magnetic turbulence. This attempt should be extended to include (global non-resonant) pressure anisotropies for both particle species, protons and electrons, in order to apply to and include mirror modes. It, however, raises the problem of identification of the possible plasma modes which could, in addition to mirror modes, be involved.

Candidates different from the mirror modes themselves are drift-waves excited in the mirror-mode boundaries and ion-cyclotron waves/ion whistlers. Similar to electron whistlers, ion-cyclotron waves propagate almost parallel to the magnetic field. Similar to what is believed of whistlers, they quasi-linearly deplete any *resonant* ion-plasma anisotropy. Under the conditions of the AMPTE IRM and Eq-S observations considered here, their frequency should be roughly $\sim 1$ Hz. They should thus appear about once per second. Inspection of the magnetic trace in Figure 3 gives no indication of their presence. The waves which we identify as the electron-mirror branch are of lower frequency.

However, any weak turbulence theory of the mirror modes should also take into account their mutual interaction. The linear dispersion relation already indicates that they are not independent. Since quasilinear theory fails in the description of their nonlinear saturated state, the nonlinear interaction of both the ion and electron mirror branches should also be taken into account in weak turbulence theory. Such a theory should lead to the final saturated state of the mirror modes which we have recently discussed in terms of basic thermodynamic theory (Treumann & Baumjohann, 2018) which, in thermodynamic equilibrium should always apply.

It can, however, not unambiguously be excluded that just these waves represent electromagnetic ion-Bernstein/ion-cyclotron waves *packets* which may have evolved nonlinearly to large amplitudes and long wavelengths in the course of weak kinetic turbulence of the ion-cyclotron wave, thereby erasing the quasilinear depletion of the resonant ion anisotropy and, in kinetic mirror turbulence contributing also to further growth of the mirror modes until they achieve their large amplitudes. Hence, here we detect a possible caveat of our investigation.

The argument against this possibility is that these ion-cyclotron waves, whether linear or nonlinear, should not be in pressure balance nor should they trap any electrons. This means they should not be related to the excitation of the observed high frequency whistlers or lion roars. However, even that argument may be weak if the ion-cyclotron wave packets locally produce enough radiation pressure to deplete plasma from their regions of maximum amplitude. Electrons reflected from those packets could then possibly locally evolve temperature anisotropies and by this cause high frequency whistlers.

Thus, one identifiable caveat remains in the possibility that ion-cyclotron waves (in both cases of AMPTE IRM and Eq-S at frequency $\sim 1$ Hz) have become involved into the weakly-turbulent evolution of the observed ion-mirror modes (or possibly also other low-frequency electromagnetic drift waves, which could be excited in the magnetic and plasma gradients at the ion-mirror boundaries). As noted above, the effect of ion-cyclotron waves is believed to quasi-linearly diminish the *resonant* part of the ion anisotropy which probably does not happen when weak kinetic turbulence takes over. In weak turbulence, the nonlinear evolution of the ion-cyclotron waves erases the quasilinear quenching and allows further growth of the ion-cyclotron waves until the waves evolve into the above mentioned wave packets of long wavelength which may become comparable to the structures identified here as the electron-mirror branch waves. This possibility, though improbable, cannot be completely excluded.

There is no final argument against the involvement of such waves other, than that ion cyclotron waves, like whistlers, are immune against pressure balance. As linear waves they do not trap any electrons which, however, might change when becoming strongly nonlinear. As long as this does not happen, it would exclude ion-cyclotron waves as source of the lion roars observed in the wave data of Figure 1.

It will be worth investigating these far reaching questions with the help of high resolution plasma, field, and wave observations from more recent spacecraft missions like MMS. It would also be worth investigating whether any electromagnetic *short wavelength* (electron) *drift modes* (cf., e.g., Gary, 1993; Treumann et al., 1991) can be detected. Those waves might, in addition to ion-cyclotron – and of course also electron-mirror branch modes themselves –, become involved into the weak turbulence of ion-mirror modes when excited at short wavelengths comparable to the plasma gradient scales in mirror modes. Their excitation is on the expense of the pressure balance. In such a case they might directly affect the macroscopic anisotropy

and contribute to weak kinetic turbulence of mirror modes while at the same time undermining their quasilinear quenching. Extension of the weak magnetic plasma turbulence theory as developed by Yoon (2007) to the inclusion of pressure anisotropy and oblique propagation could be a promising way to tackle the problem of weakly-kinetic mirror mode turbulence.

*Note added in final revision*: Ahmadi et al. (2018) in a recent study based on high time and energy-resolution electron data provided by the MMS mission (paralleled by particle-in-cell simulations), recently confirmed the generation of large amplitude high frequency whistlers (lion roars) in the flanks of the magnetosheath ion mirror modes in correlation with locally trapped resonant energetic electrons. Their results indicate the presence of localised magnetic electron traps like those provided by the electron-mirror branch. Incidentally, a statistical study of lion-roar whistlers in the magnetosheath (Giagkiozis et al., 2018) has also been published recently.

*Acknowledgement.* This work was part of a Visiting Scientist Programme at the International Space Science Institute Bern. We acknowledge the hospitality of the ISSI directorate and staff. We thank Hugo Breuillard (CNRS, Paris) for constructive comments. We also acknowledge the objections of the reviewers which required substantial revisions and extensions of the brief original manuscript.

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

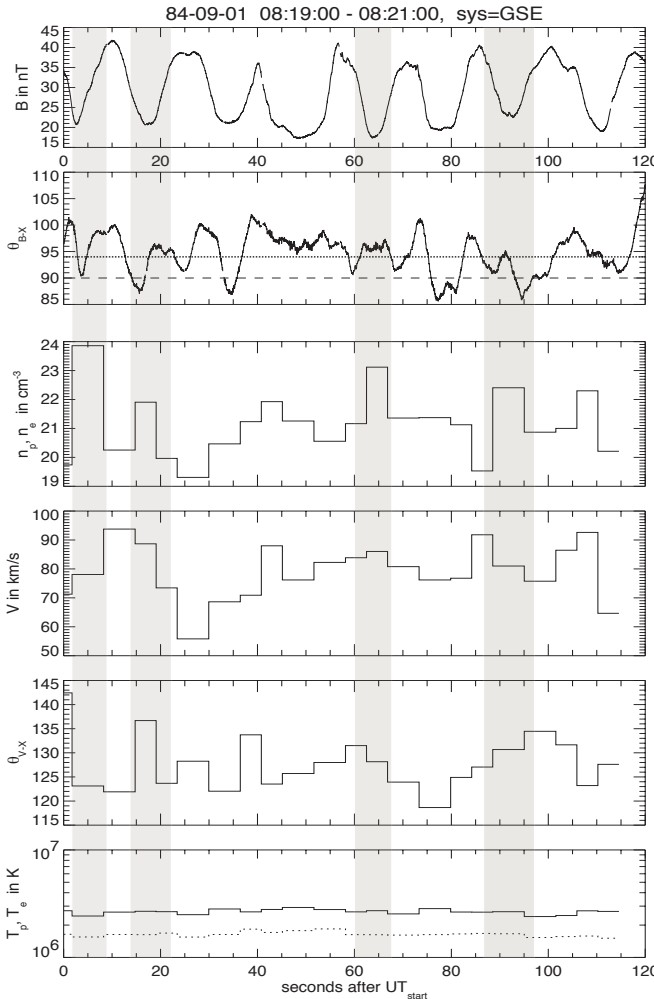

**Figure 2.** Spin-resolution ($\sim 4$ s) AMPTE-IRM plasma observations of mirror modes in the magnetosheath, lacking wave data. (Spin axis was about perpendicular to the ecliptic.) Panel 1 from top is the magnitude of the magnetic field, panel 2 angle of the magnetic vector with sun direction $X$ in GSE coordinates . The mean direction of the field is indicated by the dotted line during the entire long phase of observations of which the 2 min shown are just an excerpt (with field almost in the ecliptic and about tangential to the shock and magnetopause). Panel 3 shows the plasma density in the available spacecraft at spin resolution for a single measurement. Panels 4 and 5 give the mean velocity and direction angle of flow. Panel 6 (in logarithmic scale) is the plasma temperature, least reliable due to the resolution. Four cases of mirror troughs are shaded roughly showing the anti-correlation between magnetic field (or magnetic pressure) and density (or plasma pressure). Though this event is taken at a different occasion, it is similar to the observations in Figure 1, when no plasma data were available. One may note the small scale depressions on the average course of the magnetic trace which indicate decreases in magnitude of the magnetic field and magnetic pressure. Higher resolutions of these will be seen in the Eq-S measurements shown in the next figure. It is these depressions which we take as signatures of the electron mirror branch.

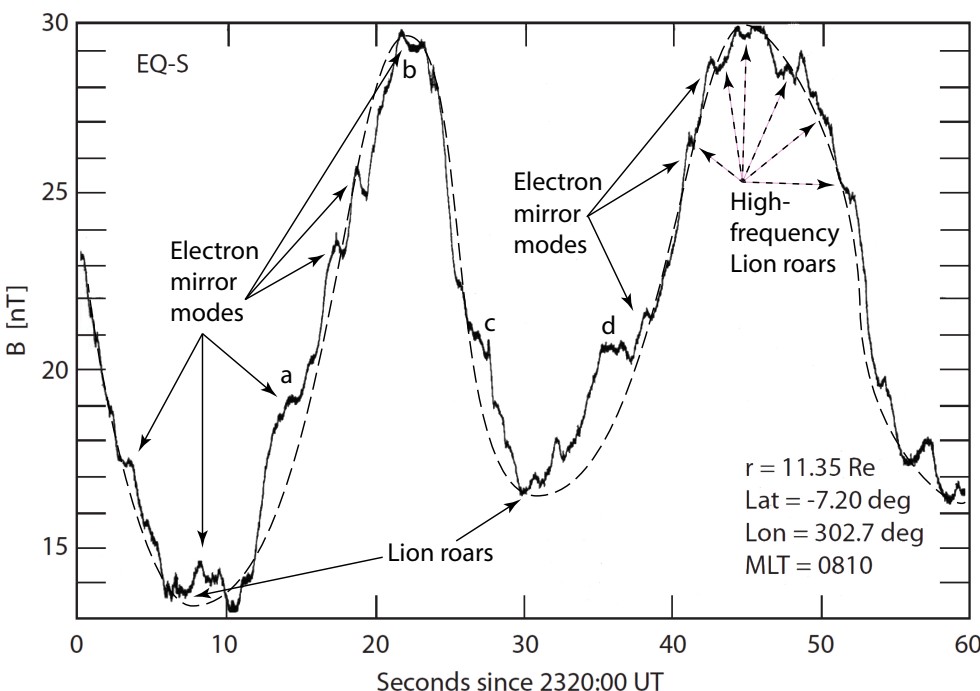

**Figure 3.** High resolution Eq-S observations of mirror modes in the magnetosheath near the magnetopause at the high sampling rate of 128 Hz. The dashed line is a low-pass filtered trace yielding a quasi-sinusoidal approximation to the ion mirror mode structure. Asymmetries are presumably caused by the obliqueness of the ion mirror mode structure in combination with the plasma flow which transports them to pass over the spacecraft. The strong modulation of their shapes on the smaller scale is produced by the superimposed small-scale electron mirror mode structure on the ion mirror mode. Signatures of low-frequency ($f \lesssim 0.1 f_{ce}$) lion roars are found in the ion mirror minima (see Baumjohann et al., 1999, for their identification, packet structure and generation). Higher-frequency lion roars concentrate in the minima of the electron mirror branch structures where they are seen as broadenings of the magnetic trace.

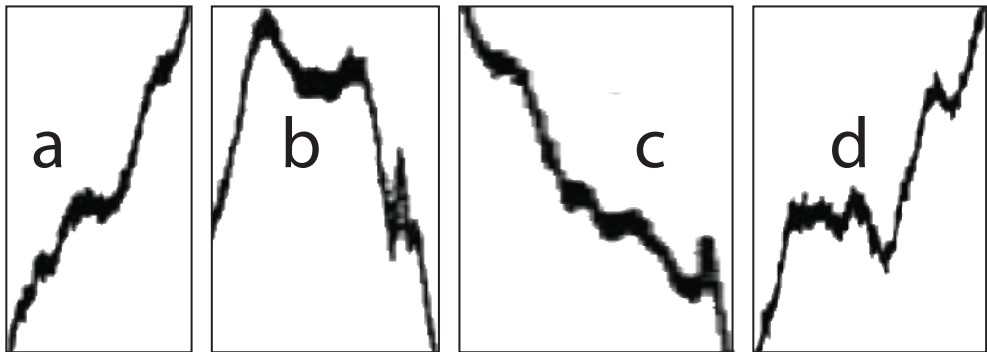

**Figure 4.** Four cases of high frequency lion roars related to electron mirror modes extracted and zoomed in from Figure 3. Cases $a, c, d$ are at the flanks of the ion mirror structures, case $b$ is on top in the first absolute magnetic maximum, all causing a broadening of the magnetic trace. At the 128 Hz sampling rate, such oscillations cannot be resolved. It allowed Baumjohann et al. (1999) to analyse the lowest frequencies ($f \lesssim 0.1 f_{ce}$ at $f_{ce} \lesssim 1$ kHz), but inhibits a similar analysis for the higher frequency lion roars. In a linear sinusoidal oscillation superimposed on the magnetic trace the oscillation would be averaged out. The broadening of the trace thus does not just indicate that lion roars are present; it also shows that the lion roars appear as nonlinear wave packets with finite averaged amplitude. Note that the lion roars in the mirror minima had already been identified as occurring in localised large amplitude non-linear whistler-wave packets!