# Peer review of "Electron mirror branch: Observational evidence from "historical" AMPTE-IRM and Equator-S measurements"

_Annales Geophysicae, 2018_

## Referee Comment (RC1) · Anonymous Referee #1 · 26 Feb 2018

Review of "Observational support for the electron mirror mode: AMPTE-IRM and Equator-S measurements in the magnetosheath" by Treumann and Baumjohann As the title implies, this paper is intended to be an observational paper demonstrating the existence of electron mirror modes. I find the observations far from convincing. There is no confirmatory plasma data showing that the plasma pressure is out of phase with the magnetic pressure. Also there is no "string" of mirror mode structures, giving clear evidence of the scale of the structures. Only a few isolated cases have been shown. I suggest that the authors find better examples for observational evidence? The authors have also not addressed why the electron anisotropy would not be absorbed by the electromagnetic whistler mode "lion roars" instead of by the electron mirror mode

instability. Thus from a theoretical point of view, electron mirror modes would not be expected. The Noreen et al. 2017 reference that the authors quote imply that electron mirror modes would be quenched at very low amplitudes and not be observable. The authors need to address this issue as well. It should be noted that Noreen et al. did not address the issue of the electron free energy being taken up by the whistler mode instability. The authors should add the many references to the ion mirror mode and lion roars, at least the fundamental ones. Some are: JGR, 81, 2261, 1976; JGR, 87, 6060, 1982; JGR, 103, 4615, 1998. On line 11, I suggest the deletion of Chandresekhar et al., 1958 because of a sign error. I suggest the addition of Phys. Fluids, 12, 2642, 1969 which establishes the nonoscillatory nature of ion mirror modes. Line 12, the authors should note that the "kinetic theory" is essentially the same as the fluid theory formulation.

---

## Referee Comment (RC2) · Anonymous Referee #2 · 1 Mar 2018

The paper shows observations of the mirror mode and electron mirror mode waves in the magnetosheath to support the previous theoretical study by Noreen et al 2017. The authors only showed the dynamic spectra or waveforms of the magnetic field, but my question is how did the authors identify the mirror, electron mirror and lion roars? Especially, for me, it is difficult to distinguish the two fluctuations in Figure 2. Since the pressure balance between the magnetic field and the ions are important for the mirror mode structures, the plot of the ion beta (and also electron beta for the electron mirror mode?) will be important. The authors explained in Lines 52-57, that the lion roars are in the whistler mode branch and mostly parallel propagation, but I cannot find the characteristic from the panels. Could the authors show more evidence for them?

[Figure]

In Figure 1, it looks the lion roars and another bursty spectra (electron mirror wave?) do not appear simultaneously. Do the authors have any comments on it? (From the difference of the energy source or other plasma conditions?) For a minor comment, please state the spec (sampling rate, dynamic range...) of the instruments that the authors used in the study.
* * *

---

## Author Comment (AC1) · 16 Mar 2018

1. I find the observations far from convincing. There is no confirmatory plasma data showing that the plasma pressure is out of phase with the magnetic pressure.

We have explicitly said, already in the original submission, that there were no plasma data in Eq-S. This had been mentioned already in Baumjohann et al 1999, and before in Lucek et al. 1998a,b to which we refer. In particular Lucek et al. have given unambiguous proof, using other arguments and observations, that the large amplitude magnetic oscillations were (ion-) mirror modes. In fact, of the sequence used in Lucek, the 1999 paper by Baumjohann took a short exerpt at high resolution (128 Hz) to demonstrate

the observation of lion roars at the bottom of the mirror troughs where the whistlers could marginally be resolved as oscillations in the magnetic field. The magnetic time resolution would, however, not be sufficient to resolve the higher frequency whistlers. All earlier observations of lion roars (see the literature) were not based on magnetic but on wave observations like in our figure 1.

What concerns AMPTE IRM, there is no need anymore to prove pressure balance. AMPTE IRM had a large record of ion mirror modes in confirmed pressure balance. The 6 min sequence shown in Fig 1 is long enough as belonging to the family of mirrors. In order to show what the reviewer demands, we nevertheless added a new figure showing plasma and magnetic data from AMPTE IRM over 6 min where the anti-correlation between ion mirror modes and plasma (density, temperature) is obvious. Three cases are indicated by shading. We dare to overload the paper with more of this.

We have also given the time resolutions. The reviewer might realize that the time resolution of 4 s spin for AMPTE IRM inhibits a clearer one-to-one anti-correlation. It could be done better statistically but for our purposes it suffices to show that there is anti-correlation. Nature was so unkind not to align the spin with the magnetic field in the mirrors such that there are only single cases which at the available spin resolution of the plasma data exhibit the anti-correlation.

What concerns Eq-S, the mirror waves shown are a high resolution 128 Hz exerpt from the full lower resolution sequences in Lucek et al. where it was shown that these are mirror modes even though no plasma data were available. Would they have been available, they could at 3 s time resolution be used marginally for demonstrating pressure balance in the ion mirror modes, but there would have been no chance to demonstrate the pressure balance in the electron mirrors. Thus the demand of the referee could anyway not have been satisfied and is in fact malevolent. All the relevant references which have published thes numbers, in order to satisfy the reviewer, have been added. We are sorry, but we do not have better examples from those measurements in this

time resolution in particular as those data have not survived or are not anymore readable from old tapes.

2. The authors have also not addressed why the electron anisotropy would not be absorbed by the electromagnetic whistler mode "lion roars" instead of by the electron mirror mode instability. Thus from a theoretical point of view, electron mirror modes would not be expected.

We are surprised! The text of the review suggests that the reviewer is not a beginner but firm in both observation and theory. However, he still is subject to the typical misunderstanding of the role of anisotropy.

In fact, the anisotropy driving any mirror modes is a fluid-macroscopic anisotropy. That which excites whistlers is the anisotropy of a small group of higher energy resonant trapped particles. The reviewer should consult Kennel and Petschek 1966 where this was quite clearly expressed. But the reviewer is excused because this misunderstanding is widely spread.

Whistlers live on the anisotropy of the energetic resonant particles (radiation belt electrons, for instance). Depletion of this anisotropy does by no means affect any possible macroscopic temperature anisotropy which drives mirror modes. For the depletion of the latter the temperature of the bulk must be changed, which the resonant particles are unable to do. They are just a few and don't do nothing on the macroscopic anisotropy.

The large 50% ion mirror amplitudes prove that depletion of anisotropy does not happen for the ion mirror mode anyway.

Lion roars are as well unable to do anything on the anisotropy which drives the electron mirror mode. Thus any quasilinear calculation misses the real effect which nobody has so far treated properly in theory. Noreen's calculation (or our own earlier 1997 quasilinear treatment of the fluid ion-mirror mode) does not say anything in this respect. It just proves that quasilinear theory does not apply to any of the observed mirror modes

because all observations demonstrate that the amplitudes exceed the quasilinear level by far. This, however, is no argument against their (or earlier) linear calculations which can be used and give reasonable results.

Concerning the references, we have included the suggestions (and a bunch of others more). Thanks for alerting us.

Needless to say that we have introduced a substantial number of explanations in the text (all in blue), added two figures: fig 2 showing the anticorrelation magnetic-plasma in AMPTE IRM, fig 4 a blow up of some regions where indications of high-frequency whistlers (lion roars) are evident from the Eq-S magnetic field trace noting that these cannot be resolved by the 128 Hz magnetic resolution which just marginally sufficed to resolve the lowest whistler frequencies at the bottom of the ion-mirror modes in Baumjohann et al 1999. The wave observations of AMPTE IRM show clear indiations of the presence of all those whistlers.

Finally, we changed the title of the paper in order to accommodate the doubts of this reviewer, since our observations and discussion is reasonable but is no direct observational proof. Some other possibilities still exist which me mention in the paper. One would be electromagnetic ion-cyclotron waves (ion whistlers) in weak kinetic turbulence which could be mistaken as electron mirrors though we argue against that possibility. We have indicated this caveat in the text. The other are short wavelength drift modes which we cannot exclude but also cannot identify. Any proper weak kinetic turbulence of ion mirror modes should identify and account for those short wavelength eletromagnetic drift waves and ion-whistlers excited in the plasma and magnetic pressure gradients inside mirror modes. These waves grow on timescales much faster than the quasilinear mirror saturation scale such that quasilinear depletion does not come into effect and the mirror mode can reach the observed large amplitudes.

**84-09-01  08:15:00 - 08:22:00,  <200 ms>,  sys=GSE**

**Fig. 1.** 6 min ampte irm plasma and magnetic field showing the anti-correlation in ion mirror modes

---

## Author Comment (AC2) · 16 Mar 2018

1. For a minor comment, please state the spec (sampling rate, dynamic range...) of the instruments that the authors used in the study.

We start with the simplest question. Magnetic observations on Eq-S were at 128 Hz sampling rate. This allowed Baumjohann et al 1999 (paper to which we refer as a basis for this investigation) to marginally (concerning sampling) resolve oscillations in the magnetic field in time at the bottom of the ion mirror modes under the conditions of a ∼30 nT main field at frequency ∼0.1 electron cyclotron frequency (the dynamic range of the magnetometer was sufficient at 0.1 nT). It did not allow resolution of whistlers

at higher frequencies above say 0.3 cyclotron. Observations of whistlers in this range have been ubiquitous when using wave-electric field instrumentation on other spacecraft (see the references for the basic papers) on which the presence of whistlers have been reasonably claimed. Since no magnetic wave observations were available for those waves, the Baumjohann 1999 paper was important to show their magnetic component thus confirming lion roars to be whistlers seen in the electric wave and the fluctuating magnetic fields. There also were the arguments given for the nature of such waves as whistlers, and even a weak resonant anisotropy in the electrons could be theoretically inferred.

2. The authors explained in Lines 52-57, that the lion roars are in the whistler mode branch and mostly parallel propagation, but I cannot find the characteristic from the panels.

From the above it is clear that higher frequency than those in Baumjohann 1999 could not be directly seen in the magnetic recordings of any, in particular not the Eq-S spacecraft. In the figure shown here, which is at the highest Eq-S time resolution (sic 128 Hz), the higher frequency > 0.3 electron cyclotron frequency whisters cannot be resolved in time. However, where the instrument could in the average detect their presence, it should observe a broadening of the magnetic trace. Inspecting the magnetic trace this is exactly what is seen and this is seen in relation to the much lower frequency magnetic oscillations overlaid on the ion mirror trace. Evidence for such higher frequency temporarily unresolved waves is therefore given in these observations. More can, however not be done.

3. Since the pressure balance between the magnetic field and the ions are important for the mirror mode structures, the plot of the ion beta (and also electron beta for the electron mirror mode?) will be important.

The reason for why not more can be done from Eq-S observations alone is that no plasma or particle measurements were available due to failure of the plasma instrument, as has been explicitly said in the text of the original submission and was learly noted in Baumjohann 1999. Thus the demand of the AR#2 (which is identical to the demand of AR#1) could not be satisfied even if we wanted. In additn, time resolution of the plasma isntrument would have been a mere ∼3 s spin which would marginally sufficed to show pressure balance with ion mirrors but would have been illusionary with electron mirrors.

4. In Figure 1, it looks the lion roars and another bursty spectra (electron mirror wave?) do not appear simultaneously.

Fig 1 is AMPTE IRM data. Here the plasma instrument had spin resolution ∼4s which is sufficient to demonstrate plasma-magnetic anticorrelation as seen in the new Fig 2 where 3 cases have been shaded. This should suffice though could be done statistically better. We consider this superficial for our purposes as it has been done in many other papers already and is well known for the ion mode. For the same reason, it would be illusionary to try to demonstrate pressure balance between electron and ion mirror modes. Thus the demand of the AR#2 (as also that of AR#1) can principally not be satisfied based on the available data.

5. The authors only showed the dynamic spectra or waveforms of the magnetic field, but my question is how did the authors identify the mirror, electron mirror and lion roars?

For the identification of ion mirrors see the above comments and papers by Lucek et al 1998a,b from Eq-S to which we refer. Concerning the distinction of We refer to the theoretical distinction between both mdoes as given from the linear calculation of Noreen. As we have said in the paper, the mirror modes are convected across the spacecraft. Hence (contrary to whistlers whose frequency is not affected by the perpendicular transport as they propagate parallel to the magnetic field and there is no Doppler shift) the frequency of the ion and electron mirror modes is low, roughly 0. Therefore their temporal scales map their spatial extension (Taylor's hypothesis!).We

thus can compare their temporal lengths. This shows that those magnetic oscillations on the flanks which we identify as electron mirrors are roughly 10 times shorter than those of the ion mirror modes in Eq-S. This corresponds almost exactly to the different ranges in the linear claculations of Noreen et al 2017. This is stong support for the electron mirror modes.

In addition, the amplitudes of the electron mirror modes (as we identify them) is much smaller than that of the ion mirror mode but much larger than those of the whistlers (lion roars) in agreement with Noreen's predictions of a factor 10 difference.

Now, this last prediction is based on a quasilinear calculation, and AR#1 has complained that the saturation amplitudes are way to large when compared with the quasilinear saturation level. This is true. But the relation between the levels is precisely what is observed.

This leads to the question, why the absolute amplitudes are so large, another factor of 10 larger than the quasilinear saturation, and this for both modes, the ion mode as well.

The answer is that quasilinear theory does not apply to the mirror modes! Mirror modes are in the weakly turbulent plasma state, where quasilinear saturation is erased by mode-mode and wave-particle interactions. The problem is that such a theory hs not yet been developed for mirror modes simply because the interacting modes have not been identified yet. We therefore propose that one of the modes participating in weak turbulence is just the electron mirror which hterefore should, in contrast to Noreen et al, not be treated quaislinearly but included into a weakly turbulent theory. Other modes can be found in electromagnetic ion-cyclotron modes and also drift-modes excited on the plasma and field gradinet in the mirror modes which may grow on the boundaries of the mirror modes and inhibit quasilinear saturation.

However, this also allows us to identify one caveat: ioncyclotron or drift waves as a possibility to replace those modes which we call electron mirrors. This can be decided

only on the basis of spacecraft data of higher plasma and field resolution.

Finally: there is a grave mnisunderstanding in the role of anisotropies in mirror modes and whistlers. These anisotropies have nothing in common with each other. Whistlers (lion roars) life from resonsnt particle anisotropies (trapped resonant electrons, a minor component of electrons), while mirrors are driven by macro-anisotropies: the temperature anisotropies of the bulk plasma. Thus evolution of whistlers on the account of resonant particles has nothing in common with the evolution of mirror modes on the expense of the temperature anisotropy. Any argument based on putting them equal is simply wrong.

6. We do not comment on the problem of the higher frequencies seen in the wave spectra of AMPTE IRM. This has been sufficiently explained already in the first version of the paper and is a little more elaborated included in the revised version.

Nevertheless, thanks to the AR#2 for forcing us to write such an extended response.

Needless to say that we have included some of these comments (in less extended form) into the revision. plus two figures which may help understanding our reasoning. We also hope that this paper will ingnite further research in the physics of mirror modes, in particular its kinetic turbulent state.

———————————————

**Fig. 1.** AMPTE IRM plasma and field data 6 min at available resolution showing the anticorrelation between B and NT. Three cases are shaded.

[Figure]

**Fig. 2.** Zoom into the magnetic trace of Eq-S showing evidence for superposition of high frequency waves related to those events we call electron mirror modes

---

## Author Response (AR2)

Reply:

This paper intended to express three points:

1. that signatures of the electron mirror branch were present already in old data but had been missed for several reason: (a) they were not expected because electrons were not believed to play any role in mirrors, (b) instrumental resolution was insufficient to resolve them;

2. that old electric wave field spectra signatures of high frequency whistlers indicated the presence of locally trapped resonant electrons, and that such trapping conditions were provided by localized depletions of the magnetic field most probably caused by the electron mirror branch;

3. that these proposals should be checked and proved with reference to high-resolution electron and field observations provided by MMS (and the MMS group to which we not do belong).

Publication of these 3 suggestions half a year ago when we had submitted this paper would have made sense.

Meanwhile, the MMS group has picked up these suggestions and has proved them correct in all points. Their  paper (Ahmadi et al. 2018) is a very important contribution to our knowledge about  mirror modes, high frequency whistlers in mirror modes and their relation to locally trapped energetic resonant electrons which show up as temporal local spikes.

Ahmadi et al.'s work is by now in print in JGR, which makes obsolete our investigation, in particular its publication at this late time. Even though proved correct, it is untimely.

We direct this reviewer to the mentioned paper where he can find out the facts and answers to his questions.

We thank him and the other reviewer (who has not responded) for their acknowledgeable efforts in handling this paper and their objections to its publication which in the end turn out to have been justified as they are overwritten by more precise measurements then those to which we had access and referred to.

Nevertheless, in order to be patient and cooperative and to satisfy the notorious wishes of this reviewer (as well as to save his valuable time), here are some brief answers to his inquiries:

1. Because of his complaints about Fig 2, Fig 2 has been replaced by a shorter excerpt from the former one in the available highest IRM plasma resolution. 4 cases of anti-correlations are indicated by shading (sorry for the reivewer who seems not to like shading, which however suffices for our modest purposes here to demonstrate kind of an anticorrelation to him). The reviewer might find those anticorrelations useful, if he wants, by inspecting the B and N traces. It not, we cannot help. We apologize if we can't do better. The old data cannot be revived anymore without

investing enormous energy. This would be nonsensical in the current situation where the proof the reviewer demands has become obsolete. No temperature anisotropies were measured by IRM, such that the traces of temperature were smooth.

Time resolution of the plasma instrument was 4 s, (we had mentioned this several times before, the reivewer apparently missed it) marginally allowing detecting the ion-mirror anticorrelation while completely inhibiting any resolution of the electron branch.

The same restriction would have applied to Eq-S where even no plasma measurements were available (for breakdown of the plasma instrument, which we noted several times before, even in the original submission). However, as also noted in the first revision, Lucek et al (1998) analyzed the whole series of Eq-S magnetic observations (of which the one presented in Fig 3 is at the highest time-resolution), to unambiguously demonstrate by other means that these oscillations were indeed ion mirrors, even though no plasma data were available. For us it completely suffices to refer to this work for a proof that we are dealing with mirrors here.

A substantial number of short time decreases occur on the traces of the magnetic field magnitude, both in IRM as Eq-S, with amplitudes orders above any magnetic fluctuation amplitudes (for thermal magnetic fluctuation spectra the reviewer, if not being firm in this field, may consult the paper of Yoon 2017). Such medium scale short time depressions of the magnetic field magnitude can hardly be explained by statistical or instrumental effects and least by magnetic fluctuations. (To make it clear: Electromagnetic signals would not change the magnetic magnitude.) They thus require compressions on the small scales. These are provided by the electron mirror branch, have been ignored before, and can only be caused by local pressure effects.

2. Concerning whistlers: The reviewer wants to see proof that the high frequency waves in the IRM spectra (or those on the Eq-S magnetic traces) are whistlers in parallel propagation. Answering this question is by now also made obsolete because the experimental answer can be found in the above cited MMS-paper.

Nevertheless, not in order to satisfy this reviewer (who ironically notes that we had provided lots of theoretical argument) but to offer some thoughts, even though no direction measurements of the waves contributing to the wave spectra was made on IRM as it contained just a single electric antenna, we note the following:

IRM had spin axis perpendicular to the ecliptic. The antenna was perpendicular to spin axis thus being in the ecliptic with signal modulated at half spin (~2 s). The electric wave field it measured was thus (about roughly) in the ecliptic. The main magnetic field direction (as seen from trace 2 in Fig 2) was almost perpendicular to the sun-Earth line (~95° in the average) though oscillating a bit as caused by the spatial organisation of the magnetic mirrors and some slight nutation of the spin axis and flapping of the 30 m long antenna. The rotating electric antenna thus measured the wave electric field in the ecliptic plane, sometimes along, sometimes perpendicular to the ambient field. However, no direction of the antenna in the ecliptic was/is known, i.e. no antenna phase is known or can be reconstructed without enormous and by now completely useless additional effort and waste of

additional time, even though this reviewer would like to see it.

The almost homogeneous weak (greenish) spectral background (below the trace of the electron cyclotron frequency) gives the spectrum of electrostatic ion-sound waves which fill the magnetosheath (this has been known for long, Rodriguez and Gurnett 1975). These waves drop out briefly when the antenna is strictly perpendicular to the magnetic field (we do not tell the reasons for this here; as they can be found in the literature). The occasional intense events (e.g. at around ~280 s) coinciding with such dropouts thus indicate the presence of the wanted perpendicular electric fields measured at such occasions. They do belong to whistlers which propagate parallel to the field. There is no other reason for them to exist unless the knowledgeable reviewer can offer some opposite argument in which we would be interested.

Whistlers in the electron branch magnetic depletions are bound to be excited when electrons are trapped in some way, like in electron mirrors (or also by electron beams to offer another explanation which shifts their observation to another unexplained cause). Since these and the direction of the antenna are uncorrelated, very little can be said otherwise about the propagation direction of the whistlers which, however, by reference to theory, propagate in a cone centered around the direction parallel to the magnetic field.

Again, the reviewer is directed to Ahmadi et al. and their MMS data for observations.

3. The reviewer complains about the lacking resolution of the waveform of the high-frequency whistlers in the Eq-S magnetic data.

He seems not to have read the brief explanation we gave in our text. In order to assist him, we repeat:

Eq-S had no wave instrumentation, just a 128 Hz time-resolution magnetometer.

In Baumjohann et al. (1999) these data had been used to resolve the magnetic waveform in the magnetic (ion mirror) minima where the resolution sufficed for mapping the waveform.

This was an absolutely major achievement in any mirror observations and theory (that paper would have deserved widespread recognition for its importance – it even identified the bulk anisotropy and range of resonant electron velocities necessary for exciting the whistlers, something which nobody had done before). This paper has been completely ignored by the self-concerned lion roar community who obviously did not grasp its brisance or tried to suppress its insight.

At those low frequencies no wave antennas work. Hence resolving the magnetic wave form and constructing its spectrum was utmostly important. It proved that these waves were indeed electromagnetic whistlers occurring in packets and being restricted to the location at the bottom of the mirror troughs.

The importance of this analysis and that paper lies not only in the waveform but also in the confirmed packet structure of the whistlers, which indicates that those

whistlers are indeed not simple sinusoidal oscillations but highly nonlinear large amplitude waves in parallel propagation being trapped in some magnetic flux tubes. The resolution of waveform and packet structure by the Eq-S magnetometer was in this case marginally possible and thus fortunate.

At the higher frequencies the Eq-S resolution was insufficient to resolve the waveform. We have said this very clearly in the original manuscript. Obviously the reviewer has again missed it. In case the higher frequency whistlers excited in the electron mirrors could be resolved, they would as well occur in packets. Such wave packets on the magnetic trace, which the Eq-S magnetometer could not resolve, are localized large amplitude waves. Their time-averaged effect contributes a rms nonzero mean fluctuation amplitude which adds to the observed magnetic magnitude. They thus appear as temporally/spatialy localised broadenings of the magnetic trace.

This is what is seen in Fig 3 and expanded for 4 cases in Fig 4.

We expect that the reviewer is capable of understanding what we have meant with this long unnecessary reply which, together with the long time of reviewing has wasted much of our time which we could have used for more important work than answering objections.

More to say is not necessary.

As for a final remark:

We are glad that our colleagues who are working on MMS were able to confirm the presence of localized whistlers and their relation to energetic electrons (which turn out to be the resonant component of a population of locally trapped electrons in the flanks of ion mirrors). Their excellent experimental, observational, and simulational work can be considered as a major advancement in our understanding of mirror structures, one of the important lowest frequency modes in magnetohydrodynamic turbulence. Our work on historical data filled just a minor gap and is by now overwritten by these observations which confirm it but at the same time render it outdated by now.

[revised manuscript text omitted]

---

## Author Response (AR3)

Reply to Reviewer 3:

We thank the reviewer for his constructive comments.

The required changes made on the original MS are shown in blue in the submitted pdf-file.
These are as follows:

Page 2, line 21 : « Here we demonstrate .. » should be replaced by « Here we suggest .. »

done (thanks very much)

Page 9, line 17 : « being somewhat shorter in parallel wavelength ... » should be replaced by « being somewhat longer in parallel wavelength ... »

done (thanks again)

Page 13, line 8 : « ...has no effect at all on the global level of anisotropy ... » this statement seems to be a little bit exaggerated and contradictory with the statement written page 12, line 15 « A low level of bulk pressure/temperature anisotropy will always be retained even quasilinearly ». The authors could attenuate and/or discuss their statement and quote for instance Sydora et al., GRL, 2007. Indeed using PIC simulations, these authors showed that the initial electron temperature anisotropy is reduced (but not cancelled) due to the parallel heating of the electrons by the wave/particle interaction.

thanks again. We reworded this sentence such that it does not contradict to the former statement. We also cited the paper by Sydora et al. which indeed fits perfectly in what we intended. The new sentence is in blue in the MS, where we wrote:

"Depletion of the latter by quasilinear saturation of the whistler instability has little effect on the global temperature anisotropy which drives mirror modes unstable; it partially quenches the emission of whistlers (or in a similar way also ion-cyclotron waves). Since, however, quasilinear saturation never completely depletes the initial resonant anisotropy, some amount of anisotropy will even in this case remain [the reader may consult the related simulations of Sydora et al{2007} which apply to the whistler case]. Such a rudimentary anisotropy which is located in velocity space around the resonant population may, if large enough, nonetheless contribute to a global anisotropy which then affects the evolution of the mirror instability."

Page 13, line 15 : replacing « exhausting » by « exhaustive » would be more appropriate and unambiguous.

done. thanks very much.

Page 14, line 24 : « for in any  weak turbulence»

changed. thanks again.